# Particulate Matter Dataset Collected with Vehicle Mounted IoT Devices in Delhi-NCR

**Ishan Nangia**     Chinmay Degwekar     Sagar Gaddam     Saswat Pujari     Ismi Abidi
Vedant Vijay          Ajay Soni          Sayan Ranu
Rijurekha Sen
Department of Computer Science, Indian Institute of Technology, Delhi

## Abstract

Air pollution is one of the biggest concerns faced by developing countries like India and the world at large. The capital of India, Delhi and the National Capital Region (NCR), sees life threatening air pollution levels. This paper presents a new Particulate Matter (PM) dataset for Delhi-NCR, which contains PM data recorded over three months from November 2020 to January 2021 over an area spanning 559 square Kms. The data has been collected using vehicle-mounted mobile sensors in collaboration with the Delhi Integrated Multi-Modal Transit System (DIMTS) buses. The 13 bus dataset has been compared with the data over the same period obtained from the pre-existing static sensors, which the buses pass by. Several Machine Learning (ML) problems have been outlined, that can be studied using this dataset, two of which, spatio-temporal interpolation and anomaly detection in IoT networks are detailed in this paper. The dataset is public at https://www.cse.iitd.ac.in/pollutiondata along with appropriate documentation. We will keep augmenting the website as new data get collected, with more buses and other pollutant sensors (SOx, NOx, COx) added to our deployment in future.

## 1   Introduction

Air pollution is a bane of modern civilization, specifically in the big cities of developing countries. *Particulate Matter (PM)* is especially dangerous, as our breathing cannot filter out the ultra fine particles. Air pollution has reached life-threatening levels in Delhi-NCR, one of the largest and most densely populated urban centers in the world. There have been studies analyzing factors affecting PM [1, 2, 3, 4, 5], but such PM measurement infrastructure are highly expensive (thousands of US Dollars per instrument). This cost-scalability trade-off is very evident from the fact that the *Central Pollution Control Board (CPCB)* and *Delhi Pollution Control Committee (DPCC)* have only 37 air pollution measurement centers in Delhi-NCR, which are thoroughly inadequate to cover the vast geography of 55,000 square Kms.

On the positive side, off-the-shelf *laser scattering* based PM sensors are now available at few tens of US Dollars (USD), along with promising work on low cost sensor calibration [6, 7, 8, 9, 10, 11] against reference grade instruments like EBAM. This paper presents a novel PM dataset for Delhi, using IoT platforms comprising such laser based PM sensors, mounted on 13 public buses in Delhi for 3 months (Nov $1^{st}$, 2020 to Jan $31^{st}$, 2021). The dataset has been cleaned and made available in a public website with proper documentation in the supplementary section. The data complements the static sensor data available from the 37 government deployed instruments in important ways.

**(a) Ground-level PM:** Our units are placed in non air-conditioned buses in the driver's cabin next to a window (see Fig. 1), to correctly capture PM values breathed in by regular commuters. This is in contrast to the static sensors deployed at high towers, where measurement captures ambient PM and not ground level PM. Our data is therefore more suitable to study commuter exposure.

**(b) Spatio-temporal Measurements:** Secondly, the static sensors record measurements *only* at the deployed locations, as shown by location icons in Fig. 2. In contrast, our instrumented buses span a large *spatio-temporal* space. To elaborate, our sensors cover a large spatial area (around 559 square

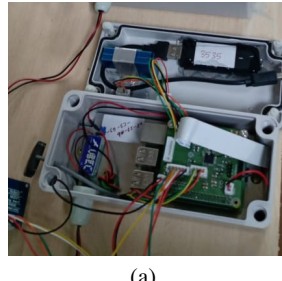 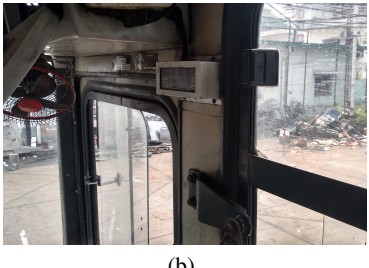 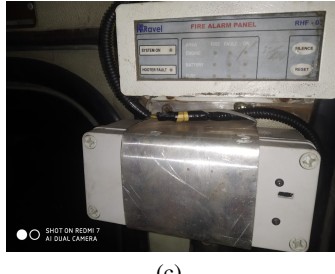

|        |        |        |
|--------|--------|--------|
| (a)    | (b)    | (c)    |

Figure 1: (a) Shows the inside of our PM measuring IoT unit. (b) Shows the mounting location in bus driver's cabin in non air-conditioned public bus. (c) Shows a mounted IoT unit in the bus.

Kms.) along three public bus routes. In addition, we deploy 3-4 buses along each route, for greater *temporal* coverage of that spatial area, totalling a 13-bus deployment. Each bus takes around 4 trips every day in each direction (source to destination and back), and we therefore obtain 24-32 trips along each route every day, getting samples almost every hour of the day along the covered routes. We compare the overall data volume by which we complement the currently available static sensor data for the same 3 months period, and also show the excellent correlation of the low-cost mobile sensor data with the expensive static sensor data, whenever an instrumented bus is co-located with a static sensor in Section 3.

**Potential Use-cases:** The proposed dataset is valuable for several downstream applications. Pollution researchers and environmental think tanks can use this dataset to study pollution trends over a given time-period. For example, PM *cluster analysis* would reveal spatial and temporal pollution hot-spots. *Correlation analysis* of PM with other factors such as green-cover, population density, wind speed, humidity, etc. may be studied by environmentalists to mine factors that influence pollution. This auxiliary data may be collected from Google satellite images for green cover detection, Google Places for neighborhood characterization such as residential areas vs. commercial, and weather data repositories for humidity, wind speed, etc. The dataset would also facilitate designing of public-awareness programs such as mobile apps to disseminate pollution levels at a much finer granularity than currently possible, recommendation of pollution-free routes (See Fig. 2(d)), identification of green-zones in a city, etc. Finally, the dataset would be invaluable for policy-decisions on deciding the optimal balance between low-cost IoT based mobile monitoring methods with high-cost sparse static sensing in developing countries. This is therefore a very important dataset from environmental sustainability stand-point, a pilot dataset which can make cost-effective scalable mobile PM monitoring a viable alternative in budget-constrained areas. We ourselves are scaling up the number of instrumented buses and the types of sensors in each bus (SOx, NOx, COx pollutants, in addition to PM), and all new data collected will be added to the same public website.

**Novel contributions compared to other mobile monitored air pollution papers:** Vehicle mounted air pollution sensing has been tried in scientific literature before [1, 2, 3, 4, 5, 12, 13, 14]. Of these, only two studies have made their datasets publicly available, namely [13] and [14] which collected data in Ontario, Canada and Zurich, Switzerland respectively. Compared to these two publicly available datasets, our dataset has very different distribution of PM values. This is understandable as Delhi, the area under our study is an air pollution hotspot, whereas Zurich and Ontario have negligible PM levels. Also, the Zurich dataset does not include PM values, while the Canada dataset does. We perform a detailed statistical comparison of our dataset vs. the Canada PM dataset in Appendix. It shows the higher spatial and temporal variations in PM, as well as much higher mean PM values. Regarding generalizability of the dataset, Delhi-NCR pollution is likely to correlate well with other regions in the Indo-Gangetic plain, and portions of Pakistan [15, 16], which is one of the most densely densely populated regions of the world.

In addition to be of value to environmental researchers as discussed above, modeling the far more characteristic spatial and temporal patterns in this dataset can also be interesting to Machine Learning (ML) researchers. We demonstrate a sample modeling exercise for spatio-temporal interpolation with several ML baselines on our dataset in Section 4. A second important distinction is the developing country constraint of our deployment, that necessitated a low cost IoT network deployment using 4G cellular connections on public buses. This is in contrast with thousands of USD worth instrumented research vehicles used in prior studies, using dedicated broadband network. Our dataset from a live 3 month IoT network deployment can thus enable novel research directions for ML researchers, in

automated anomaly detection in IoT networks (discussed in detail in Section 5). Finally, the GPS part of the dataset can be interesting for other ML problems like automated detection of traffic congestion in non-laned traffic, as Delhi is also notorious for traffic congestion, and our buses cover the same routes every day in peak and non-peak hours.

## 2 Data Collected, Cleaned and Released

### 2.1 Vehicle mounted sensor deployment in Delhi public buses

We worked with Aerogram, our IIT Delhi incubated startup involved in designing and building low-cost Air Quality Monitoring (AQM) devices. The AQM device we build for our buses is "EzioMotiv"[1][2] which is a robust spatio-temporal data collection device allowing the measurement of particulate matter (PM1.0, PM2.5, PM10.0), along with GPS values. The PM sensors of EzioMotiv are tested and calibrated rigorously in-house with standard devices such as Beta Attenuation Monitor (BAM). In Fig. 2, we show the government deployed static sensors installed in and around our area of interest. Furthermore, we overlay a heatmap of the bus trajectories for different days on the same spatial zone. We compare the values across our mobile and the government installed static sensors in Section 3. Our IoT unit, mounting position inside the bus and a mounted unit are shown in Fig. 1.

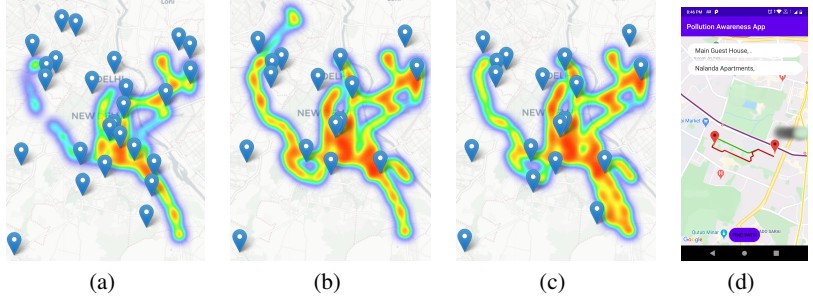

| (a) | (b) | (c) | (d) |

Figure 2: Heatmap of the buses attached with our devices. Reference grade static sensors in our area of study for which we could find data using the OpenAQ platform are shown with landmark icons. (a) shows the data for $26^{th}$ November 2020 (b) shows the data for $11^{th}$ December 2020 (c) shows the data for $1^{st}$ January 2021. These maps are meant to show a glimpse of the variation in both the bus trajectories and in the data available for static sensors. (d) shows a pollution aware route app that we have built for our university campus (similar can be built at city-scale using data like ours.)

### 2.2 Data cleaning

**Particulate Matter data:** We drop duplicate entries, process the date-time stamps and convert them into *Indian Standard Time (IST)* format. We then drop entries with PM1, PM2.5 or PM10 values above 2000 or below 0. These unrealistically high PM values occur when our sensor is directly in front of a source of smoke or dust, such as the exhaust of a vehicle (as confirmed with in-lab experiments). Further, we label any and all PM values recorded above 1000 and below 20 as outliers, based on data analysis and discussion with domain experts. We replace outliers by a median-based moving average of 10 entries.

**GPS data:** Our incorrect GPS coordinates are either 0 or negative and the ones that are above 0 but are not in Delhi. For the latter, we take a bounding box of Delhi and drop all entries outside the bounding box. The occurrence of the former anomaly is natural in our case. Since our sensors are attached inside the metallic body of the bus, obtaining GPS coordinates through satellite connection becomes tough sometimes because of loss of connection under bridges, when travelling underground etc. To deal with these, we formulate a problem as follow: suppose we know the location tuples $(x_1, y_1)$ and $(x_2, y_2)$ for a bus at timestamps $t_1$ and $t_2$, we wish to find its possible location at time $t \in (t_1, t_2)$. We solve this using *Map Matching* [17, 18, 19] or *Linear Interpolation*, both of which give good accuracy, with the latter having lower running times. We detail the two methods and evaluation in the supplementary section.

---

[1]https://aerogram.in/products/eziomotiv/
[2]https://www.evelta.com/content/datasheets/203-PMS7003.pdf

### 2.3 Released dataset

Our collected and cleaned dataset can be found at http://pollutiondata.cse.iitd.ac.in/. The data is organized filewise for each day, each file containing values for up to 13 sensors deployed in buses. Not all 13 buses are active on all days of the 3 month deployment period (Nov $1^{st}$, 2020 to Jan $31^{st}$, 2021). DeviceID, GPS location, time-stamp, PM10, PM2.5 and PM1 values are listed in each row of a file. We also include the anomalies manually found in the dataset as ground truth files, so that ML researchers can use that as a starting point for ML based automated anomaly detection in IoT networks. Detailed documentation of the dataset is available at the website and in our supplementary material, which we will keep updated as new data gets collected, cleaned and added to the website.

## 3 Dataset Utility for Environmentalists

To monitor pollution levels in the city, at least 37 static sensors have been installed by governmental institutes spanning the entire city of Delhi. These sensors are located at the top of high towers to get precise recordings of ambient pollution values, not affected by local sources. Our mobile sensors, on the other hand, are installed in the bus driver's cabin near a window to measures the ground level pollution that daily commuters breathe in. Environmentalists can therefore use these complementary datasets to understand PM at tower heights using static sensor data, as well as near ground using our mobile sensor data. In addition to the difference in heights where PM values are measured, we compare below the quantity and quality of our mobile monitoring data with static sensor data. This exercise allows us to quantify our dataset's utility for environmentalists, who already have access to the static sensor dataset[3].

### 3.1 Data Volume: Mobile Sensors vs. Static Sensors

Our mobile sensors spatially cover more area compared to static sensors. As shown in Fig 2(a)-(c), the static sensors do *point* measurements where the location icons are, while the buses cover the entire region shown as location heatmaps. However, the static sensors get values for the point location at all times, whereas mobile sensors collect some values at a location and move. Thus, we compare the overall temporal volume of data collected by the two sensor types to compare their data quantities.

For static sensors, an active hour must have at least one value reported, as their sampling frequency is low. For mobile sensors, a device is said to be active in an hour if it sends at least 800 data points in that hour. This indicates the mobile device being active for at least 40 minutes in the hour with ideal sampling rate of 20 data points in a minute. Figure 3 shows cumulative active hours per device from $16^{th}$ November 2020 to $30^{th}$ January 2021 for static and mobile sensors. Each curve belongs to a different instrument. When a curve is flatter and more parallel to $x$-axis, then the number of active hours for that instrument is less on that particular day. Similarly, if the curve is steeper and is more parallel w.r.t $y$-axis, then there are more active hours that day.

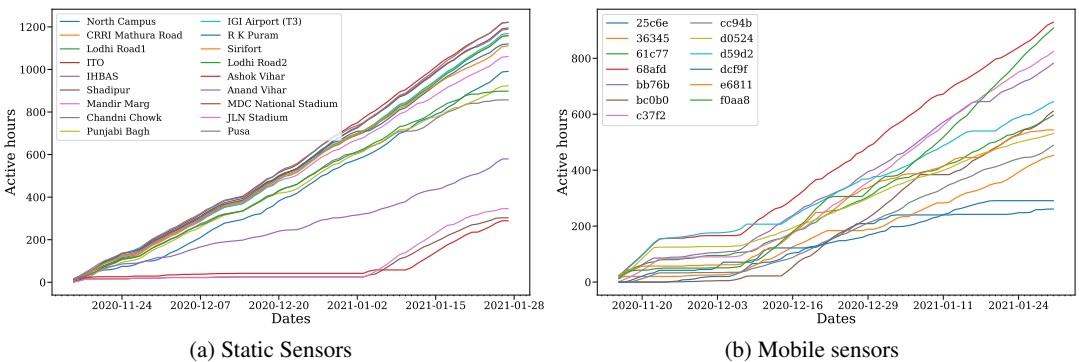

(a) Static Sensors                      (b) Mobile sensors

Figure 3: Cumulative active hours for static sensors (left) and mobile sensors (right) between $16^{th}$ November 2020 to $30^{th}$ January 2021. Color viewing recommended.

From the plots, we observe that most of the static sensors are active for most days though they have much less values per hour (1 to 5 readings). The mobile sensors are anomalous initially, when the

---

[3]https://openaq.org

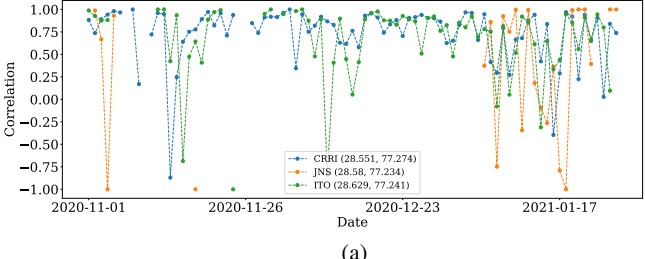 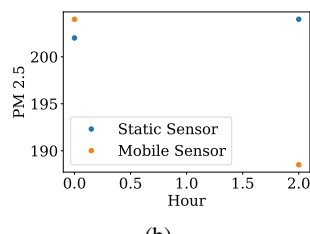

|          (a)          |          (b)          |

Figure 4: (a) Correlation between hourly averages of static and mobile sensors from $1^{st}$ November 2020 to $30^{th}$ January 2021. The three line plots correspond to three different locations where we consistently found the mobile sensors to be passing by the static sensor. We see that most of the instances have a positive correlation indicating reliability of our low-cost sensors. (b) Shows a common example of negative correlation between the static and mobile PM values. The correlation here is -1. The PM values recorded by both the sensors are extremely close in magnitude and thus the negative correlation can be ignored.

mobile network was being setup, until around $10^{th}$ of December as indicated by the lower slopes of the curves before that date. After Dec $10^{th}$, the mobile sensors become ideal for the remaining days, with more active hours per day and also with many more samples per hour (800 to 1200 readings) compared to static sensors. Thus, the mobile dataset adds significant quantity of data for the environmentalists to analyze, once the mobile network became stable since Dec $10^{th}$.

### 3.2 Quality: Mobile Sensor Measured PM Vs. Static Sensor Data

We additionally compare the quality of our mobile PM measurements with static sensor measurements in nearby locations. The number of static sensors reporting data varies a lot from day to day, though we are unsure whether this is a hardware or a software issue on the government's maintained static sensing infrastructure. We have data from 18 sensors available on $12^{th}$ November 2020. However, on $26^{th}$ December 2020 the number of reporting sensors increase to 36. On $1^{st}$ January 2021, we get data from 24 sensors. Fig. 2 shows the static sensors in our region of study for which data was available on that day for the above three days. To compare our measurements, we first locate the mobile sensors that were close (less than or equal to a distance of 150m), to any static sensor. We found three static sensors satisfying this criteria, which were installed at CRRI Mathura Road, Delhi, Jawaharlal Nehru Stadium, Delhi and ITO, Delhi respectively (referred to as CRRI, JNS and ITO respectively in Fig 4).

We compute the correlation between the hourly mean of all PM2.5 values recorded by a static sensor and its nearby mobile sensors. Since the PM values should roughly move from one hour to the other hour in the same direction (increase or decrease), we expect to see a high positive correlation between both the hourly averages. Fig. 4 (a) shows the daily correlation values of all three locations from $1^{st}$ November 2020 to $30^{th}$ January 2021. Region corresponding to JNS lacks some values, as static sensor data was unavailable for comparison in that period. We observe a high correlation across most days. Differences in raw PM2.5 values between two types of sensors are caused by a variety of reasons including the difference in heights they have been installed at, the difference in the amount of exposure of the sensor to direct smoke and dirt, the difference in measurement technique and the averaging procedure introduced by us to compare the sensors.

The three static sensors were strategically selected to ensure that the accuracy of all mobile sensors can be evaluated. To elaborate, the three static sensors are located in three diverse but highly busy intersections. Hence, a large portion of the mobile sensors get spatio-temporally close to at least one of the static sensors. To give some concrete statistics, on average 78% of the mobile sensors get calibrated against once of the three static sensors per day. Across the entire time-duration, all mobile sensors have been calibrated at least over 8 days against one of the static sensors. The distribution of number of mobile sensors that were calibrated against a static sensor per day is provided in Fig. 13a in Appendix. Furthermore, in Fig. 13b, we provide the distribution of number of days that a mobile sensor is calibrated against one of the static sensors. Overall, this data provides confidence that the mobile sensors have been compared against a substantial number of static sensor readings.

We also found 15 instances where the correlation was found to be negative. Fig 4 (b) shows a common example of the type of instances we found to give out negative correlation. Correlation does not take into account how close in magnitude the recorded PM values are. These were cases where PM values

were extremely close in magnitude and thus, their trend became trivial. Overall, the correlation with static sensors is remarkable, given the positioning and measurement technique differences. We had 177 total correlation values out of which 103 were found to be significant at $\alpha = 0.05$. The mean correlation of this subset was 0.85. Only one significant negative correlation was found. This will hopefully encourage environmentalists to consider vehicle-mounted sensing as a low-cost scalable PM measurement alternative, to complement expensive static sensors in other polluted cities.

## 4   Dataset Utility for ML Researchers: Spatio-temporal Interpolation

Our dataset is a potential gold-mine for Machine Learning researchers. § 1 discussed several potential ML problems. In this section, we present a concrete case-study of spatio-temporal interpolation.

Spatio-temporal Interpolation is a widely explored ML research problem [20, 21, 22, 23, 24, 25, 26, 27, 28, 29]. Since our mobile sensors record PM data only on a subspace of the entire spatio-temporal space, the problem of PM interpolation on the uncovered space is a natural consequence. We study this problem: given a training set of tuples of the form $\langle date, time, latitude, longitude, pm2.5 \rangle$, learn a model to predict the PM2.5 value on an unseen spatio-temporal point $p = (date, time, latitude, longitude)$. Since we focus on the interpolation problem, we assume the unseen point $p \in \mathcal{B}$, where $B$ is the bounding box over all spatio-temporal points seen in the train set. The baseline methods we use for interpolation on our dataset are described in brief below, with more details in the supplementary section.

**[1] Gaussian Process Regression (GPR):** A Gaussian process (GP) [30] is a collection of random variables, any finite number of which jointly follow a Gaussian distribution. GPR is a Bayesian regression approach which assumes a GP prior over functions. This is a commonly used tool [23, 24] for spatio-temporal interpolation.

**[2] Variation Gaussian Process Regression (Variational GPR):** Variational GPR [31] uses a variational inducing point framework to scale GPR for bigger datasets by using subtle and smart approximations.

**[3] GraphSAGE:** GraphSAGE is a framework for inductive representation learning on large graphs [32] which generates low-dimensional vector representations that can be then used for classification, regression etc. on previously unseen data. We use a model heavily inspired by GraphSAGE. We have not included the TRAIN_RMSE of this model in Table 1 because of long running times.

**[4] Graph-based Weighted Mean (Meaner):** This is a custom weighted mean method where we fit a graph to the data and based on the edges and their weights, we use a weighted mean to predict the pollution values for all the train and test points. TRAIN_RMSE of this model couldn't be reported due to long running times. The concept of epochs doesn't apply to this model since it doesn't require training. Hence, only TEST_RMSE$_{FULL}$ and TRAIN_TIME$_{FULL}$ have been reported.

**[5] Artificial Neural Network (ANN):** Fully connected ANNs are common interpolation baselines [23]. We use a weight initialization technique proposed by [33] along with Adam optimization [34] with exponentially decaying learning rate.

To evaluate each method, we take five common randomly chosen days from our data and separate out the data hour-wise from 9:00AM to 9:00PM. We then split the data randomly into train and test using a ratio of 80:20 for each day. We further split the train data into train and validation sets using the same ratio. For learning the embeddings, all models are fitted onto a downsized train dataset, where PM of nearby locations and times are averaged to reduce data granularity. Then the learnt model is used upon the original train and test datasets to get the train root mean square error (TRAIN_RMSE) and the test root mean square error (TEST_RMSE) [35], compared to the ground truth PM values. We also measure the time taken to get the final model (TRAIN_TIME) (in seconds) and provide information on if the method outputs variance values (VAR). Further these metrics are calculated for two different cases. The first is the 100 epoch case where each model is trained only for 100 epochs. The metrics pertaining to this case are denoted by writing 100 in the subscript. The other is the full training case where each model is trained till an early stopping criterion is met. The metrics pertaining to this case are denoted by writing $FULL$ in the subscript. The first case was tested to check whether models can be used in a real-life deployment with streaming data as training the models to achieve a decent error rate should be possible in a short period of time.

Table 1: Spatio-temporal Interpolation Baselines

| Models | Metrics | | | | | | |
|---|---|---|---|---|---|---|---|
| | $\text{TRAIN\_RMSE}_{100}$ | $\text{TEST\_RMSE}_{100}$ | $\text{TRAIN\_TIME}_{100}$ | $\text{TRAIN\_RMSE}_{FULL}$ | $\text{TEST\_RMSE}_{FULL}$ | $\text{TRAIN\_TIME}_{FULL}$ | VAR |
| GPR | **32.66** | **32.89** | 92.56 | **31.35** | 31.77 | 1091.81 | ✓ |
| Variational GPR | 35.75 | 35.82 | 159.12 | 31.38 | **31.73** | 1464.50 | ✓ |
| GraphSAGE | × | 34.85 | 5531.13 | × | 34.60 | 5619.81 | × |
| Meaner | × | × | × | × | 34.52 | 2005.47 | × |
| ANN | 40.09 | 41.03 | **2.51** | 33.45 | 33.77 | **41.73** | × |

The results are documented in Table 1. For each metric, the value of the model with the best score are marked in bold. We observe from $\text{TRAIN\_RMSE}_{100}$, $\text{TEST\_RMSE}_{100}$ and $\text{TRAIN\_TIME}_{100}$ that most of these models can be used easily in a streaming fashion. GPR, Variational GPR and ANN all take really less time to train and get a decent RMSE, as compared to their convergence RMSEs. However, both the graph models are exceptions here. They achieve low RMSEs but take significant time to do so. This happens primarily because of a major overhead: the time taken up by the graph building process which takes in the train and validation datasets as inputs and returns a graph fitted on the datasets. Making the edges for each node consumes a lot of time. The more the number of train and validation nodes, the more the time it takes to build up the edges of the graph. Similarly, depending on how huge a test dataset is, creating edges between train and test nodes doesn't scale well. We observe that $\text{TRAIN\_TIME}_{FULL}$ depends only on the time to preprocess the dataset and train the final model. Pre-processing takes the most time but training is extremely fast, especially when compared to the other methods like Variational GPR. ANN on the other hand take the least time and thus can be quickly used to produce interpolation maps for deployment purposes. GPR is a fast method too but has a limitation on the size of the dataset it can process. Thus, when using this on larger datasets, it can run into memory-based problems. Since the model is fixed for Meaner and getting the predictions on a given graph doesn't take much time, its $\text{TRAIN\_TIME}_{FULL}$ indicates mainly the time taken to preprocess and build the graph on our dataset.

In terms of $\text{TRAIN\_RMSE}_{FULL}$ and $\text{TEST\_RMSE}_{FULL}$, the fitting of the gaussian process-based methods is the best. Thus, these become the de facto methods to use when working with large scale historical data. The difference between the RMSEs of the other models is not huge. Their performances are more or less similar. The graph-based methods rank the last in terms of the quality of the fit to the dataset. The final column in Table 1 shows one of the main advantages of GPR and Variational GPR. Both methods output a variance value with each prediction. This value shows how confident the model is about its prediction and is thus extremely useful for a number of purposes including model-based predictive control [36], comparing space-filling designs [37] and active learning [38]. None of the other models output this value.

Further optimisation could potentially decrease the TRAIN_TIME for each method. Our methods had to downsize the training data since it was taking too much time to train the model. However, even after downsizing the dataset, the models could achieve a relatively really low RMSE on both the train and test datasets. This data therefore contains some redundancies which makes it possible to use a compressed version of the data to get low RMSEs. ML methods exploiting such redundancies to be more sample-efficient, can therefore be examined using this dataset.

We additionally run each interpolation model on three different random splits of the five randomly chosen dates and present the average results with standard deviations across the 15 runs in Table 7 in Appendix. While the absolute numbers change slightly between Table 1 and Table 7 due to averaging, overall the relative trends across the different models hold. We also compare the Delhi vs. Canada dataset interpolation difficulty using the ANN method in Appendix. As visible in Table 10, the ANN interpolation errors in Canada is 3 times lower than in Delhi. This is a direct consequence of the Delhi dataset being (1) much more heterogeneous with a high spatio-temporal variance and (2) significantly higher mean pollution levels. This shows that our dataset is more challenging to model than other available datasets in the same domain, and therefore should be useful to ML researchers exploring novel spatio-temporal interpoltaion methods.

The purpose of this section was to show that this dataset can be used to explore spatio-temporal interpolation methods by ML researchers. We used five models of slightly different flavors: ANN and Graph Neural Networks (in GraphSAGE) and Gaussian Process Regression (very widely used by air pollution researchers). Our goal here is not to get best possible modeling accuracy, but more to show that our dataset is useful for this kind of modeling problem. We will explore more models [39, 40, 41] in future, which can potentially work better for this dataset.

# 5 Dataset Utility for ML Researchers: Anomaly Detection in IoT Networks

This dataset has been created using a novel IoT network with low cost sensor platform, deployed in public buses in a developing country, the first of its kind. There are many points of faults — sensors can be faulty, internet connection can be shaky, buses might be down .... the faults can affect the quantity of data as well as quality. Detecting such anomalies for quick fixes is a necessity. In this paper, we apply statistical analysis to detect anomalies, which involves many heuristics with manually tuned thresholds. Our findings can serve as anomaly ground truth for this dataset. Automating this process with ML based methods (instead of manually tuned thresholds) can open up new avenues of anomaly detection in mobile and IoT networks. ML researchers can try and automate the fault detection process using our dataset and the ground truth anomalies. They can also modify our released code, to change our empirical thresholds for more or less aggressive anomaly definition. Additionally unsupervised learning methods perhaps would change the anomalies detected manually by us. We describe next the different anomaly metrics that we compute on the dataset using statistical analysis and empirically determined thresholds.

**Anomaly metric 1: Samples recorded per minute:** This metric checks for faulty devices which might be sampling more or less than expected rate. Fig. 5a shows ideal samples collected per minute should be around 20. If it deviates too much, that device is anomalous. The amount of deviation allowed is calculated statistically by observing the distributions for several days. Our algorithm (detailed in the supplementary section) finds the upper bounds and lower bounds of the median ($\Theta_{50}^L$, $\Theta_{50}^U$), $25^{th}$ percentile ($\Theta_{25}^L$, $\Theta_{25}^U$) and $75^{th}$ percentile ($\Theta_{75}^L$, $\Theta_{75}^U$) of the expected distribution. Anomaly is reported if any two of the three bounds are violated.

**Anomaly metric 2: Number of minutes each device is active in an hour:** A device can be active

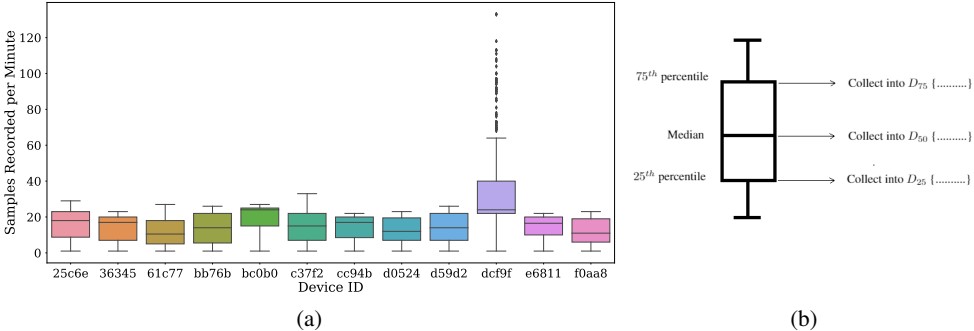

(a)                                                    (b)

Figure 5: (a)Sampling rate i.e. number of samples recorded per minute for 3.12.2020. This helps us in finding out if any device isn't sampling properly.(b) Illustration of a sample box plot and process of collecting median, $25^{th}$ percentile and $75^{th}$ percentile in metric 1 and 2

for all the 60 minutes of an hour or less based on time of the day/lunch break, stoppage at bus depots etc. So again we tried plotting the box plots for the distributions across the days and devices. We observe that ideally device should be active for 60 minutes of an hour, if the bus was taking a trip in that hour. So we used the same technique used in Metric 1: find the upper bounds and lower bounds of the median ($\Theta_{50}^L$, $\Theta_{50}^U$), $25^{th}$ percentile ($\Theta_{25}^L$, $\Theta_{25}^U$) and $75^{th}$ percentile ($\Theta_{75}^L$, $\Theta_{75}^U$) of the expected distribution. Anomaly is reported if any two of the three bounds are violated.

**Anomaly metric 3 : Number of active hours in a day:** An active hour for a particular sensor is any hour in which the sensor sends at least a fixed number($\gamma$) of samples. The number of active hours should ideally be greater than a threshold value($\tau$). But it is hard to fix one $\tau$ across all sensors, as different buses have different schedules and frequencies, which also can change over time. This is shown by the bar plots of number of active hours in Fig 6.

So we define $\tau$ or ideal number of active hours for every sensor as the maximum of 10 active hours and $15^{th}$ percentile of the sensor's previous 15 days' active hours. This ensures that $\tau$ is appropriately chosen for every sensor depending on its particular bus's recent schedule. If the number of active hours for a sensor on any day doesn't satisfy the threshold ($\tau$), it is reported as anomalous for the day.

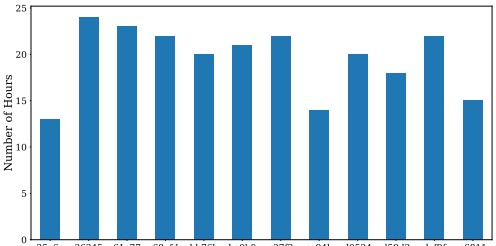 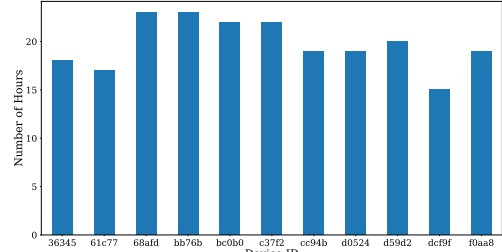

Figure 6: Number of active hours for different devices on two different days as shown by two different plots.

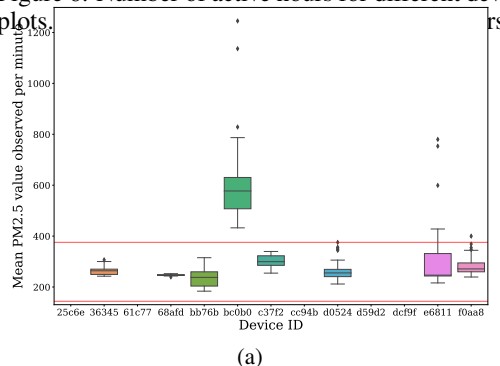 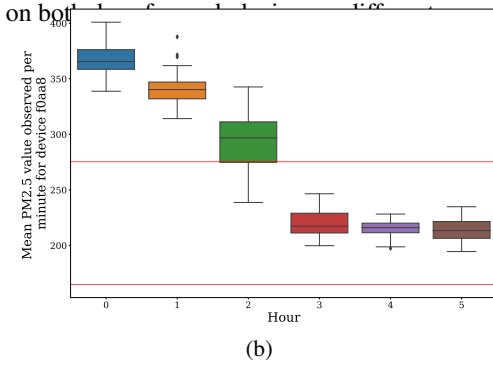

(a)                                                      (b)

Figure 7: (a) shows a box plot of PM value distributions of various devices on 2020-12-20 4:00 AM IST. Red lines indicate the majority PM range. Device bc0b0 is flagged as it is out of majority PM range while device e6811 is flagged as its IQR exceeds max_IQR. They will be declared as anomaly if they show this behaviour in at least two more hours. (b) shows PM value distributions of the device f0aa8 on 2021-01-16. The device is reported as anomaly as its PM value measurements are highly varying across several hours.

**Anomaly metric 4: Samples recorded per region:** It is important to check whether daily around same number of data points are collected or not in an area. This metric detects situations where bus may not complete its scheduled trip due to mechanical breakdowns or high traffic resulting in less recording of data points in some areas. We divided the area covered by buses into 16 square regions. Given total number of data points collected in each region on a day, a region is reported as anomalous if its value deviates from past seven days average of that region by at least $\delta\%$. The value of $\delta$ is calculated by observing the data of several days.

**Anomaly metric 5: Inter-sensor PM values variation:** Ideally the PM values measured by different sensors should lie in a close range if the measurements were carried out at the same location and time. Every night from 0 AM IST to 5 AM IST all the buses remain parked at the same bus depot. We have used the PM value data from this time period to find devices whose PM 2.5 value measurements deviate from the general PM value trend of majority devices. For each hour, we have a box plot describing PM value distributions of all the devices as shown in Fig. 7a. Let $\Theta_{25}$ and $\Theta_{75}$ represents 25th and 75th percentile of distribution of PM values of a device during an hour. Interquartile range (IQR) is defined as ($\Theta_{75}$-$\Theta_{25}$). Given a box plot, a device is flagged for possible anomalous behaviour if it's IQR is very high (e.g. device e6811 in Fig. 7a). In order to define how much IQR should be considered high to be flagged, we define a threshold max_IQR which is set as 90 percentile of all the IQRs of all the devices in training set. Secondly, if a box (middle 50% data) of a device in the plot varies in a range different than the range of other devices then also the device is flagged (e.g. device bc0b0 in Fig 7a). To find such anomalies, we first find a range in which boxes of majority devices lie, then all those devices which are out of this range are flagged. A parameter called 'buffer' is defined statistically based on training data for finding the range. Given a PM value distribution of an hour, we iteratively calculate candidate range as [$\Theta_{25}$-buffer, $\Theta_{75}$+buffer] for each device. The candidate range which contains the boxes of maximum number of devices is considered as the final range. The devices whose box does not fit completely in this range are flagged for that hour. Finally a device is reported as anomalous if its get flagged for at least three hours in a day.

**Anomaly metric 6: Intra-sensor PM values variation:** Similar to the above metric, intra sensor analysis verifies that the variation in a device's PM value recordings across consecutive hours is not

very high. Given a device's PM value recordings during 0 AM IST to 5 AM IST, it is flagged for further checks if IQR of the device during any hour is greater than max_IQR or if at least three boxes lie out of majority PM range. The majority PM range is the PM value range which contains the maximum number of boxes computed similarly as described in the above metric except that the value of buffer here is computed based on intra senor PM value distributions. Finally a device is reported as anomalous only if its get flagged for at least three hours in a day. Figure 7b shows one such anomaly.

We detail the heuristics for computing the above six anomaly metrics, the thresholds and summary statistics of all anomalies found, in the supplementary section and the website. The anomalies found in the paper were cross-checked with the platform vendor Aerogram and the deployment partner, the public bus company DIMTS, for correctness and usefulness. All cases on inter-sensor and intra-sensor variations (metrics 5 and 6) were caused by local electrical maintenance work in a particular bus at the depot, whose sensor readings deviated from other buses in the depot. Lack of samples per minute or per hour (metrics 1 and 2) are helping to understand 4G networking issues. Finally the metrics for active hours per day and spatial coverage consistency (metrics 3 and 4) are helping to gain insights on unpredictable public bus behavior in Delhi, especially during Covid-19 induced lockdowns, where bus schedules and routes are seeing significant variations. Thus all these anomalies are highly important to gain insights about a live IoT network deployment. Defining these metrics and the multiple thresholds for them has been cumbersome, and more automated ML methods using this dataset and our findings as ground-truth, will be immensely valuable.

## 6   Conclusion and Future Work

We release a PM dataset for Delhi in this paper, collected between Nov $1^{st}$, 2020 and Jan $31^{st}$, 2021 using IoT devices deployed in the driver's cabin in 13 public buses. The data has been cleaned and correlated with the government deployed static sensors, whenever our bus mounted unit has come in the vicinity of a static sensor. Environmental researchers can use this complementary dataset, to understand PM exposure at ground levels, as well as PM temporal variations over days, weeks and spatial variations across locations. We also outline several ML research problems that can use this dataset, and discuss in depth two of these. For the *spatio-temporal interpolation problem*, we present several baseline models' performance on the dataset. For the *anomaly detection in IoT network* problem, we present 6 anomaly metrics that we manually define and detect in our dataset using heuristics and empirically analyzed thresholds. These detected anomalies, along with the code, are released to be potentially used as ground truth, for ML based automated anomaly detection.

Table 2: Cost in INR and availability of gas sensors

| Pollutants | vendor1 | vendor2 | vendor3 |
|---|---|---|---|
| SO2 | 17,337 | 8,500 | 10,600 |
| NO | 14,600 | 8,500 | - |
| NO2 | 10,618 | 8,500 | 10,530 |
| CO | - | 8,500 | 4,450 |
| CO2 | - | - | 2,910 |
| O3 | - | - | 4,288.28 |

We are currently extending the sensor deployment to more buses and and other vehicle fleets (shared cabs, delivery agencies' two-wheelers etc.), and augmenting our IoT unit with gas sensors. We will keep augmenting the dataset (with comparisons with static sensors and anomalies detected, as done here) when more data gets collected. The main bottleneck in adding gas sensors is the relatively higher costs of these sensors (shown in Table 2) compared to PM sensors, which cost only 2,500 INR. Thus we are carefully considering budget availability, so that at least some of our instruments can have a richer set of sensors. We are also collecting and analyzing auxiliary datasets that affect PM: road traffic congestion data (from this dataset's bus GPS traces, Google Traffic APIs), green cover vs. built area information (from satellite image datasets), residential areas vs. commercial areas (from Google Places and OpenStreetMap POI datasets), so that our measured PM values can be correlated at fine-grained spatio-temporal scales with the auxiliary data for factor analysis. Finally, we use GPR-based methods and vanilla neural networks (which do not output variance). Our MSE loss metric is outlier sensitive. As our dataset contains outliers, a more fair baseline to compare would be to use likelihood estimation methods and Bayesian neural networks [39, 40, 41], that tolerate outliers better. We will explore these interpolation methods in future.

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
