# OpenReview forum: "Particulate Matter Dataset Collected with Vehicle Mounted IoT Devices in Delhi-NCR"
_NeurIPS.cc/2021/Track/Datasets_and_Benchmarks/Round1 — Submitted to NeurIPS 2021 Datasets and Benchmarks Track (Round 1)_

### Official Review · Reviewer_KeDp · 2021-06-17
**Large-scale dataset but no quantitative comparison with other publicly available benchmarks**

**Rating:** 6
**Confidence:** 4

**Strengths:**

1. The dataset is large in both spatial and temporal scales.
2. A wide range of use cases by environmentalists and ML researchers are discussed and analyzed.
3. The GPS information is provided and the comparison between static and mobile sensors are given.

**Weaknesses:**

1. There is no statistical comparison with previous PM datasets such as [13] and [14], which makes it difficult to justify the difficulty and significance of the proposed benchmark.
2. The IoT units were installed in the public buses for data collection. There is no explanation on how they gained the permission. The ethical issues might be a concern.
3. There is no clear introduction to the IoT devices used and analysis on their accuracy.
4. The audience is small compared to other CV and NLP based datasets.
5. It might be better to provide the ground truth of test set separately or create a public submission system/leaderboard for fair comparison.
6. What is the plan for maintenance and licensing?

**Additional Feedback:**

Please refer to my feedback above.

**Clarity:**

The paper is well written with clear motivation. I would suggest to create a separate table to compare the statistics with existing public benchmarks.

**Correctness:**

The dataset is constructed in a sound way. The evaluation methods make sense. Multiple experimental designs are provided.

**Documentation:**

The data collection and organization are documented in sufficient detail. But there is no description of the plan for licensing and maintenance.

**Ethics:**

As mentioned in the weaknesses, the IoT units were installed in the public buses for data collection. There is no explanation on how they gained the permission. The ethical issues might be a concern.

**Relation To Prior Work:**

As mentioned in the summary, compared with other mobile monitored air pollution works, the proposed dataset has different distribution of PM values as the area under their study is an air pollution hotspot. Due to local constraint, the data were collected using 4G cellular connections on public buses.

**Summary And Contributions:**

A particulate matter (PM) dataset is collected with vehicle mounted IoT devices in Delhi, India. Compared with other mobile monitored air pollution works, the proposed dataset has different distribution of PM values as the area under their study is an air pollution hotspot. Due to local constraint, the data were collected using 4G cellular connections on public buses. The dataset is also large in scale and the spatio-temporal characteristics are available. A number of use cases like sample modeling and anomaly detection are discussed.

---

> ### Author Response · Authors · 2021-07-11
> **Response to "There is no statistical comparison with previous PM datasets such as [13] and [14], which makes it difficult to justify the difficulty and significance of the proposed benchmark."**
>
> We have now included a detailed comparison with both the Canada and Zurich datasets. The details of our comparison is available at https://www.cse.iitd.ac.in/pollutiondata/novelty. The key differentiating factors of our dataset with the two existing ones are as follows:
>
> 1. Zurich dataset does not record PM values and hence is not a substitute for our dataset
>
> 2. Scale: Our dataset contains 12.5 million samples, whereas the only other dataset that records PM pollution (Canada) has only ~46k points. In other words, the proposed dataset is more than 250 times larger
>
> 3. Uniqueness: The released dataset has a significantly different characteristic compared to Canada. To be more specific, while the mean PM 2.5 and PM10 values in Canada are ~15 and ~12 respectively, in our dataset the mean values are ~208 and ~226, which is close to 15 times higher. Furthermore, the standard deviations in our dataset are 10 times higher (114 and 123 in Delhi NCR against 13 and 9 in Canada), which indicates the difficulty of pollution monitoring in the Delhi-NCR region.
>
> 4. In the above URL, we have also added spatial maps of the distribution of PM values in Delhi-NCR and Hamilton, Canada. The high variance and the higher PM levels are clearly indicative of the different modelling challenges that the proposed dataset brings. Furthermore, the high variance may also allow us to perform more find-grained analysis on how pollution-levels affect health, impact of green cover on PM levels, etc., which would not be feasible on the Canada dataset due to its homogeneity.
>
> 5. Temporal Coverage: Our comparison with the Canada dataset also reveals that there are several temporal windows when no or a very low number of PM values have been recorded in the Canada dataset. In contrast, in our Delhi-NCR dataset we have at least on sample across every minute on the timespan that our dataset covers.
>
> We will include the above discussion in our paper should it get accepted.

---

> > ### Comment · Reviewer_KeDp · 2021-07-13
> > **Regarding the statistical comparison with previous PM datasets**
> >
> > I appreciate the responses from the authors.
> >
> > To clarify, I would recommend to show the statistical comparison, e.g., the performance of the same methods on different datasets in quantitative measurements. This will help demonstrate the challenge and difficulty of this newly proposed dataset.

---

> > > ### Author Response · Authors · 2021-07-14
> > > **Performance Comparison on Canada Vs Delhi**
> > >
> > > Please find below the results obtained using the ANN method [1] (also discussed in Section 4 of our paper). As visible, the error in Canada is 3 times lower than in Delhi (our data), and this is a direct consequence of the Delhi dataset being (1) much more heterogeneous with a high spatio-temporal variance and (2) significantly higher mean pollution levels.
> > >
> > > Metric			          Delhi	Canada
> > >
> > > Training RMSE	         32.9	         10.79
> > >
> > > Test RMSE		          34.07	11.94
> > >
> > > We are currently consolidating all the changes we have made to address the comments raised by the reviewers. In the next few hours, we will update the paper and the supplementary information. Among our changes, we have added an exclusive section in our appendix (Section A.7) comparing the Canada dataset with ours. The above comparison is added in Section A.7.6.
> > >
> > > We appreciate the constructive feedback received from the reviewers.
> > >
> > > [1] Alexeef SE, Roy A, Shan J, Liu X, Messier K, Apte JS, Portier C, Sidney S, and van denEeden SK. High-resolution mapping of traffic related air pollution with google street view carsand incidence of cardiovascular events within neighborhoods in oakland.Environmental Health,2018.

---

> > > > ### Comment · Reviewer_KeDp · 2021-07-14
> > > > **Questions on performance comparison between Canada and Delhi**
> > > >
> > > > Thanks to the authors for quickly getting the new experimental results comparing the Canada benchmark with their proposed one.
> > > >
> > > > However, the reported performance on the proposed dataset (Training RMSE 32.9 and Test RMSE 34.07) is contradicted to Table 1 in the original paper (Training RMSE 33.45 and Test RMSE 33.77).
> > > >
> > > > It is also surprising that the convergence of ANN is so fast. Please describe the implementation details (computer settings, programming language, etc.). And please ensure all the methods are executed in the same environment (CPU vs. GPU). It is also not clear whether the network has fully converged.
> > > >
> > > > Besides, all the methods in Table 1 are traditional approaches and cannot be considered as the state-of-the-art for this problem. Please show the experimental results using the latest models, probably from the Papers with Code.

---

> ### Author Response · Authors · 2021-07-12
> **Response to "There is no clear introduction to the IoT devices used and analysis on their accuracy."**
>
> Accuracy of Data:
>
> Section 3.2 discusses the analysis of measurement accuracy of our mobile sensors by comparing against static sensor readings (gold standard). Section A.5 in the supplementary discusses the volume and comprehensiveness of the data against which the accuracy calibration has been done. https://www.cse.iitd.ac.in/pollutiondata/sensor_calibration contains further calibration data and correlation analysis.
>
>
> Introduction to Sensors:
>
> Section 2.1 introduces the sensors used for our study. In the same section, in footnote, we link (https://aerogram.in/products/eziomotiv/) to the technical specifications page of the sensors that includes more information measurement capabilities, power draw, synchronization options, physical specifications, storage options, etc.

---

### Official Review · Reviewer_6Ab6 · 2021-07-04
**PM Dataset for Indian NCR**

**Rating:** 4
**Confidence:** 3
**Correctness:** Please see the weaknesses section
**Clarity:** Yes, the paper is clear and well-orga…

**Strengths:**

- Public PM datasets are available for only two relatively pollution-free cities.
- The use of bus-mounted sensors instead of static sensors captures better spatio-temporal data compared to static sensors. They also capture the ground statistics instead of the statistics at the top of the tower.
- Experiments bench-marking different models in Table 3 are also lovely.
- The paper is well-organized.

**Weaknesses:**

- One of the major concerns of the dataset is that it only provides PM statistics. Table 3 of [1] contains several different pollutants.  Since the authors intend to add other pollutants such as SOx, NOx, and COx in the future, I feel that this paper submits an ongoing work, and the work is not complete. Therefore, I advise the authors to get the data for all other pollutants and then re-submit the paper.

- Do authors plan to add the ozone measurement as well as done in [2]?

- Comparison of statistics with other public and non-public air pollution datasets based on the number of pollutants covered, area mapped, sensors/vehicles used, and monitoring days in tabular form would be a nice addition .

- Would you mind adding a plot that compares the pollution levels across different datasets similar to Table 3 of [1]?

- Another potential problem is the number of static towers available to compare with. The authors only found three towers to compare the bus measurements, which is too small to conclude.

- Table 1 only compares baselines over the new NCR dataset. The addition of baselines for publicly available air pollution datasets would be nice. This experiment also lets us infer whether the performance of models depends on the pollution levels.

- The authors only compare the GPR-based methods with the vanilla neural networks (which do not output variance) with the MSE loss (which is outlier sensitive). As the dataset contains outliers, a more fair baseline to compare would be to use likelihood estimation methods [3,4] that tolerate outliers better and Bayesian neural networks.

- The authors do not report the variation across the runs in Table 1 by running the experiment multiple times.

References-
[1] A mobile air pollution monitoring dataset, Adams et al., Data 2019

[2] Sensing the air we breathe, Li et al., AAAI 2012

[3] What uncertainties do we need in computer vision, Kendall et al., NeurIPS 2017

[4] LUVLi face alignment, Kumar et al., CVPR 2020

**Additional Feedback:**

Since the authors intend to add other pollutants such as SOx, NOx, and COx in the future, I feel that this paper submits an ongoing work, and the work is not complete. Therefore, I advise the authors to get the data for all other pollutants and then re-submit the paper.

**Documentation:**

Yes, there are all sufficient details.

**Ethics:**

I do not think their are ethical concerns. The authors do not collect any personally information in the dataset. They only provide location of public buses and PM levels which should be OK.

**Relation To Prior Work:**

No. Comparison of statistics with other public and non-public air pollution datasets based on the number of pollutants covered, area mapped, sensors/vehicles used, and monitoring days in tabular form is missing.

**Summary And Contributions:**

The paper introduces a Particulate Matter (PM) dataset for Indian National Captial Region recorded over three months over a large area using bus-mounted sensors. The dataset captures the spatio-temporal PM trends in one of the most populated regions when the pollution peaks. The paper also benchmarks a few baselines over this dataset. The dataset is super-relevant to the developing world.

---

> ### Author Response · Authors · 2021-07-11
> **Response to "Comparison of statistics with other public and non-public air pollution datasets based on the number of pollutants covered, area mapped, sensors/vehicles used, and monitoring days in tabular form would be a nice addition ."**
>
> The below response clarifies the concerns raised with respect to:
>
> * Comparison of statistics with other public and non-public air pollution datasets based on the number of pollutants covered, area mapped, sensors/vehicles used, and monitoring days in tabular form would be a nice addition .
>
> * Would you mind adding a plot that compares the pollution levels across different datasets similar to Table 3 of [1]?
>
> -------------------------
>
> We have now included a detailed comparison with both the Canada and Zurich datasets. The details of our comparison is available at https://www.cse.iitd.ac.in/pollutiondata/novelty. The key differentiating factors of our dataset with the two existing ones are as follows:
>
> 1. Zurich dataset does not record PM values and hence is not a substitute for our dataset
>
> 2. Scale: Our dataset contains 12.5 million samples, whereas the only other dataset that records PM pollution (Canada) has only ~46k points. In other words, the proposed dataset is more than 250 times larger
>
> 3. Uniqueness: The released dataset has a significantly different characteristic compared to Canada. To be more specific, while the mean PM 2.5 and PM10 values in Canada are ~15 and ~12 respectively, in our dataset the mean values are ~208 and ~226, which is close to 15 times higher. Furthermore, the standard deviations in our dataset are 10 times higher (114 and 123 in Delhi NCR against 13 and 9 in Canada), which indicates the difficulty of pollution monitoring in the Delhi-NCR region.
>
> 4. In the above URL, we have also added spatial maps of the distribution of PM values in Delhi-NCR and Hamilton, Canada. The high variance and the higher PM levels are clearly indicative of the different modelling challenges that the proposed dataset brings. Furthermore, the high variance may also allow us to perform more find-grained analysis on how pollution-levels affect health, impact of green cover on PM levels, etc., which would not be feasible on the Canada dataset due to its homogeneity.
>
> 5. Temporal Coverage: Our comparison with the Canada dataset also reveals that there are several temporal windows when no or a very low number of PM values have been recorded in the Canada dataset. In contrast, in our Delhi-NCR dataset we have at least on sample across every minute on the timespan that our dataset covers.
>
> We will add the above discussion to our paper as well.

---

> ### Author Response · Authors · 2021-07-11
> **Response to "One of the major concerns of the dataset is that it only provides PM statistics. Table 3 of [1] contains several different pollutants. Since the authors intend to add other pollutants such as SOx, NOx, and COx in the future, I feel that this paper submits an ongoing work, and the work is not complete. Therefore, I advise the authors to get the data for all other pollutants and then re-submit the paper."**
>
> The below comment is in response to:
>
> * One of the major concerns of the dataset is that it only provides PM statistics. Table 3 of [1] contains several different pollutants. Since the authors intend to add other pollutants such as SOx, NOx, and COx in the future, I feel that this paper submits an ongoing work, and the work is not complete. Therefore, I advise the authors to get the data for all other pollutants and then re-submit the paper.
>
> * Do authors plan to add the ozone measurement as well as done in [2]?
>
> -------------------------------------------
>
> We agree that one shortcoming of our work is that it is only limited to PM values due to budget constraints. In https://www.cse.iitd.ac.in/pollutiondata/novelty, we present the cost of sensors to measure these additional pollution factors including ozone. In the future, we do hope to add them to our repository.
>
> Having said that, PM values are arguably the most dangerous factors for human lungs [13]. Consequently, a large body of work exists that measure and develop models only on PM values [1-12]. Hence, we believe PM values alone would be useful for environmentalists. We also emphasize that the scale, high mean pollution levels, and dramatic spatio-temporal variance imparts unique characteristics to our dataset that is not available elsewhere.
>
> --------------------------------
> References:
>
> [1] Do, Tien Huu, et al. "Matrix completion with variational graph autoencoders: Application in hyperlocal air quality inference." ICASSP 2019-2019 IEEE International Conference on Acoustics, Speech and Signal Processing (ICASSP). IEEE, 2019.
>
> [2] Lin, Y., Mago, N., Gao, Y., Li, Y., Chiang, Y. Y., Shahabi, C., & Ambite, J. L. (2018, November). Exploiting spatiotemporal patterns for accurate air quality forecasting using deep learning. In Proceedings of the 26th ACM SIGSPATIAL international conference on advances in geographic information systems (pp. 359-368).
>
> [3] Zhang, Yong, and Wulin Jiang. "Pollution characteristics and influencing factors of atmospheric particulate matter (PM2. 5) in Chang-Zhu-Tan area." IOP Conference Series: Earth and Environmental Science. Vol. 108. No. 4. IOP Publishing, 2018.
>
> [4] Li, X., Y. J. Feng, and H. Y. Liang. "The impact of meteorological factors on PM2. 5 variations in Hong Kong." IOP Conference Series: Earth and Environmental Science. Vol. 78. No. 1. IOP Publishing, 2017.
>
> [5] Li, Jinchao, et al. "Research on influential factors of PM2. 5 within the beijing-tianjin-hebei region in China." Discrete Dynamics in Nature and Society 2018 (2018).
>
> [6] Yang, Qianqian, et al. "The relationships between PM2. 5 and meteorological factors in China: seasonal and regional variations." International journal of environmental research and public health 14.12 (2017): 1510.
>
> [7] Li, Junmin, and Luping Wang. "The research of PM2. 5 concentrations model based on regression calculation model." AIP conference Proceedings. Vol. 1794. No. 1. AIP Publishing LLC, 2017.
>
> [8] Kumar, M., et al. "Wintertime characteristics of aerosols at middle Indo-Gangetic Plain: Impacts of regional meteorology and long range transport." Atmospheric Environment 104 (2015): 162-175.
>
> [9] Srinivas, Reka, et al. "Sensitivity of online coupled model to extreme pollution event over a mega city Delhi." Atmospheric Pollution Research 7.1 (2016): 25-30.
>
> [10] Feng, Jialiang, et al. "Concentrations, seasonal and diurnal variations of black carbon in PM2. 5 in Shanghai, China." Atmospheric research 147 (2014): 1-9.
>
> [11] Tai, Amos PK, Loretta J. Mickley, and Daniel J. Jacob. "Correlations between fine particulate matter (PM2. 5) and meteorological variables in the United States: Implications for the sensitivity of PM2. 5 to climate change." Atmospheric environment 44.32 (2010): 3976-3984.
>
> [12] Trivedi, Dinesh Kumar, Kaushar Ali, and Gufran Beig. "Impact of meteorological parameters on the development of fine and coarse particles over Delhi." Science of the Total Environment 478 (2014): 175-183.
>
> [13] https://www.epa.gov/pm-pollution/health-and-environmental-effects-particulate-matter-pm, https://www.health.ny.gov/environmental/indoors/air/pmq_a.htm#:~:text=Exposure%20to%20fine%20particles%20can,as%20asthma%20and%20heart%20disease

---

> ### Author Response · Authors · 2021-07-12
> **Response to "Another potential problem is the number of static towers available to compare with. The authors only found three towers to compare the bus measurements, which is too small to conclude."**
>
> The three static sensors were strategically selected to ensure that the accuracy of *all* mobile sensors can be evaluated. To elaborate, the three static sensors are located in three diverse but highly busy intersections. Hence, a large portion of the mobile sensors get spatio-temporally close to at least one of the static sensors. To give some concrete statistics, on average 78% of the mobile sensors get calibrated against once of the three static sensors per day. Across the entire time-duration, *all* mobile sensors have been calibrated at least over 8 days against one of the static sensors.
>
> The distribution of number of mobile sensors that were calibrated against a static sensor per day is provided in Fig. 13a in appendix. Furthermore, in Fig. 13b, we provide the distribution of number of days that a mobile sensor is calibrated against one of the static sensors. Overall, this data provides confidence that the mobile sensors have been compared against a substantial number of static sensor readings (gold standard).
>
> The above discussion has been included in Appendix A.5 of the updated supplementary information. We also discuss why majority of the other static sensors are not a desirable choice for calibration since they rarely get spatio-temporally close (located close to a mobile sensory and both record PM values around the same time) to each other.

---

### Official Review · Reviewer_c8qT · 2021-07-04
**A new dataset of particular matter**

**Rating:** 6
**Confidence:** 3
**Correctness:** The construction process is introduce…
**Clarity:** Yes, the paper is well written mostly.

**Strengths:**

Overall, I think this paper studies an interesting problem that connects machine learning with air pollution.

The strengths include:

+ A new dataset recording the data of particular matter in Delhi-NCR. The collection process is new compared with using static sensors, whose methodology could be further adopted in other countries/areas.

+ A detailed analysis of the dataset, including the comparisons with static sensors, quality of the data, etc.

+ The contribution could be significant since it uses machine learning to solve social problem.

+ Two new tasks are introduced using this dataset.

**Weaknesses:**

Although this paper proposes to study two tasks, i.e., spatio-temporal interpolation and anomaly detection in IoT networks, using the new dataset, these tasks seem to be less useful in practice. It may need to predict the PM for other locations, but I might think this dataset can have bigger impacts and solve larger problems than the existing ones. For example, can the data be analyzed with other information (weather, transportation information, etc.) to find important factors that affect the air pollution level. And can the findings suggest ways to control air pollution? I think the longstanding goal is to use machine learning techniques for making the air pollution better.

Another issue is about the generalizability of the proposed datasets/method to other scenarios, e.g., time and location. Since it is collected within a limited period (three month) and only in Delhi-NCR, can it be representive enough?

**Additional Feedback:**

I suggest the authors to further consider more tasks about this dataset to improve the potential impacts.

**Documentation:**

Yes, the documentation and presentation are clear.

**Ethics:**

I'm not sure about the ethical issues. In my mind, I do not see any ethical issues.

**Relation To Prior Work:**

The related work is not introduced. I'm not sure about the comparison with related work. Is there any other dataset doing a similar thing?

**Summary And Contributions:**

This paper collected a new dataset of particular matter in Delhi-NCR, which can be used to study air pollution. Such data is collected using vehicle-mounted mobile sensors on buses rather than static sensors.

The contributions of this paper include:

1. The motivation of this paper is good. Air pollution is a big problem in developing countries and raised concerns all over the world. Studying this issue from a machine learning perspective is good and interesting.

2. The data collected process is introduced clearly. The data has been analyzed and compared with that obtained by static sensors.

3. The new dataset introduced new machine learning tasks, including spatio-temporal interpolation and anomaly detection in IoT networks, which are studied in this work.

---

> ### Author Response · Authors · 2021-07-12
> **Response to Reviewer's comments**
>
> We agree that more impactful problems can be studied on this dataset. We discuss several of these problems in the "Potential Use Cases" paragraph in Section 1. While we are currently studying some these discussed problems, they are not mature enough for publication yet. Given that the primary focus of NeurIPS 2021 Datasets and Benchmarks Track is on the dataset itself, we feel that releasing the dataset would be useful for the community in general.
>
> Regarding generalizability of the dataset, although Delhi-NCR is a highly populated and industrial area and thus the pollution levels in this specific region is one of the highest in the world, it is likely to correlate well with other regions in the Indo-Gangetic plain, and portions of Pakistan [1,2].
>
> [1] Toward cleaner air for a billion Indians, Joshua S. Apte, Pallavi Pant, Proceedings of the National Academy of Sciences May 2019, 116 (22) 10614-10616; DOI: 10.1073/pnas.1905458116
> [2] Ojha, N., Sharma, A., Kumar, M., Girach, I., Ansari, T. U., Sharma, S. K., ... & Gunthe, S. S. (2020). On the widespread enhancement in fine particulate matter across the Indo-Gangetic Plain towards winter. Scientific reports, 10(1), 1-9.

---

> > ### Comment · Reviewer_c8qT · 2021-07-20
> > **Thanks for the Response**
> >
> > Thanks for the responses to my questions. I'd like to see more interesting applications of this dataset in the future.

---

### Author Response · Authors · 2021-07-15
**Paper and supplementary materials revised**

Thanks to all the reviewers for their time and valuable feedback. We have incorporated these in the revised paper (one additional page with new content in blue font) and the supplementary material (3 new sections in the appendix, now 12 pages compared to the original 7 pages). We have also updated our website with these new information.

New work done with added results (in summary):
(a) Comparison with prior PM dataset from Canada, along with ANN interpolation method run on both datasets. Establishes the spatio-temporal richness of this data, along with modeling challenges for ML researchers from higher variability. New page in website https://www.cse.iitd.ac.in/pollutiondata/novelty. New section in Appendix (A.7). Pointers in main paper for appendix in pages 2 and 7.

(b) More accuracy analysis of our dataset, and quantifying exactly what fraction of our bus mounted sensors go near static sensors for comparison. New page in website https://www.cse.iitd.ac.in/pollutiondata/sensor_calibration. New section in Appendix (A.5). Pointer to appendix added in page 5 of main paper.

(c) Multiple runs for the interpolation methods and reporting average values with standard deviations across runs. New section in Appendix (A.6). Pointer to appendix added in page 7 of main paper.

(d) Ethical concerns about permissions taken for the public bus deployment. Uploaded all certification and government permission related documents in new page in website https://www.cse.iitd.ac.in/pollutiondata/Ethical_documents.

We acknowledge the shortcomings of the current work in page 10 of main paper, which we are actively working on to improve in future. These include adding expensive gas sensors in our instruments. This is a budget bottleneck for us to balance scaling to many buses, vs. adding more expensive gas sensors to each instrument, which we are trying to resolve. But PM itself is of tremendous interest to environmentalists and health experts as discussed in our Jul 11 comment. Our future work further includes analyzing and comparing PM with auxiliary data (traffic, weather, green cover etc.) and try more interpolation models, especially Bayesian techniques.

Finally, we agree that the potential audience of our dataset is smaller than computer vision or NLP. However, PM data, arguably, has more important consequences on human lives than databases such as Imagenet (CV) or Enron (NLP).  Furthermore, it could also be argued that the ML audience for PM data is smaller due to lack of high-quality datasets itself. So, we also hope that our dataset would help the community grow. Finally, we have outlined several ML research problems that are of value to spatio-temporal machine learning or anomaly detection, and thus the scope of this dataset is not limited to just environmentalists.

---

### Decision · Program_Chairs · 2021-07-26

**Decision:**

Reject

**Comment:**

A strength of the paper is that it provides a new engaging dataset for a pertinent societal problem. However, the dataset is limited and feels like too much of a work in progress. For example, the dataset only covers three months of time for a very specific region. Moreover, the authors have future plans for other associated data to be linked and additional transports to include the sensors. Thus, it looks promising but premature. We encourage the authors to include a solid comprehensive set of data/sensors for a longer period of time and resubmit the work.